# *Bacillus amyloliquefaciens*-*9* Reduces Somatic Cell Count and Modifies Fecal Microbiota in Lactating Goats

**DOI:** 10.3390/md19080404

**Published:** 2021-07-21

**Authors:** Yongtao Li, Nannan Jiang, Wenying Zhang, Zhengbing Lv, Jianxin Liu, Hengbo Shi

**Affiliations:** 1Key Laboratory of Molecular Animal Nutrition, Ministry of Education, Institute of Dairy Science, College of Animal Sciences, Zhejiang University, Hangzhou 310015, China; liyongtao@zju.edu.cn (Y.L.); jiangnannan@mails.zstu.edu.cn (N.J.); zhangwenying@mails.zstu.edu.cn (W.Z.); liujx@zju.edu.cn (J.L.); 2College of Life Sciences and Medicine, Zhejiang Sci-Tech University, Hangzhou 310018, China; zhengbingl@zstu.edu.cn

**Keywords:** probiotics, subclinical mastitis, gut microbiome, immune response

## Abstract

Subclinical mastitis is one of the major problems affecting dairy animals’ productivity and is classified based on milk somatic cell counts (SCC). Previous data showed that marine-derived *Bacillus amyloliquefaciens*-*9* (GB-9) improved the immunity and the nonspecific immune defense system of the body. In this study, the potential role of GB-9 in improving subclinical mastitis was assessed with *Radix Tetrastigmae* (RT) as a positive control in subclinical mastitis Saanen dairy goats. The current data showed that GB-9 and RT significantly reduced the SCC in dairy goats. After being fed with GB-9 or RT, the decreased concentrations of malondialdehyde, IgA, IgM, IL-2, IL-4, and IL-6 were observed. The amplicon sequencing analysis of fecal samples revealed that GB-9 significantly altered the bacterial community. *Bacteroides* and *Phascolarctobacterium* were the major genera that respond to GB-9 feeding. The correlation analysis using weighted gene co-expression network analysis showed a MePink module was most associated with the serum concentrations of immunoglobulin and interleukin. The MePink module contained 89 OTUs. The feeding of GB-9 in decreasing the SCC was associated with the altered abundance of *Bacteroides*, which was correlated with the concentrations of immunoglobulins and chemokines. Collectively, the current data suggested that marine-derived GB-9 could be a helpful probiotic to control subclinical mastitis.

## 1. Introduction

Mastitis is a major problem in dairy herds worldwide and causes considerable economic losses [1,2]. Intramammary infections with various pathogenic bacteria often cause mastitis. The mammary gland attracts immune cells, such as white blood cells, to clear the bacteria, resulting in an elevation in the milk somatic cell count (SCC). Subclinical mastitis accounts for the majority of mastitis cases and is characterized by an increase in SCC (value > 200 × 10^4^/mL for goat milk) without clinical signs in the appearance of milk or the udder, as well as decreased milk production and altered milk composition [3]. Antibiotic treatment is the mainstream method used to treat subclinical mastitis in the dairy industry [4]. Antibiotic therapy significantly reduces the risk of having mastitis but may lead to the decline of dairy cow quality and formation of antibiotic residues and antibiotic resistance [5,6]. With increasing pressure to reduce antibiotic usage in dairy farms, alternative antibiotic treatment methods are strongly needed [7].

In the past years, the role of diverse gastrointestinal microbes in the well-being of ruminants has been investigated [3]. The microbiota of the intestine interacts with the host immune system, thereby influencing the development of the disease [8,9]. Previous data in humans and cows reported that the gut microbiota affects the development of mastitis [10]. The striking divergence in mastitis-associated intestinal microbiota is conservative among lactating mammals [2,11]. An alternative cure of mastitis is the use of probiotics to restore intestinal microbiota function [2]. In this regard, the modification of the gut microbiota profiling by probiotics or pharmaceutical molecules can possibly alter the status of the mammary gland, although the mechanism remains unclear.

*Bacillus amyloliquefaciens*-*9* (GB-9) function is associated with the alteration of microbial community structure [12]. A previous study revealed that *Bacillus thuringiensis* reduced the risk of subclinical mastitis of dairy goats by inhibiting microorganisms [10], indicating the potential role of probiotics in mastitis treatment. *Bacillus amyloliquefaciens-9* (GB-9) was originally isolated from the intestinal tract of a white spotted bamboo shark (*Chiloscyllium plagiosum*) [13]. GB-9 showed broad-spectrum antibacterial properties and can secrete lipopeptides, glycosidase, and other substances with antibacterial properties, which inhibit the growth of Gram-negative and Gram-positive bacteria [14,15,16]. In vivo data of fish showed that GB-9 improved the immunity and the nonspecific immune defense system [13,17]. *Radix Tetrastigmae* (RT) is an anti-inflammatory herbal medicine used traditionally. RT contains various antibacterial substances, including flavonoids, phenolic acids, and polysaccharides [18,19], which interfere with glycolysis and gluconeogenesis in bacteria and can be used to treat bovine mastitis [20,21]. Previous data suggested that flavonoids are among the main components with anti-tumor and antioxidant properties [19,22,23]. In this study, RT was used as a positive control to assess the role of GB-9 in improving subclinical mastitis.

In this study, it was hypothesized that feeding GB-9 would reduce the SCC and alter the gut microbiota profiling. Dairy goats with subclinical mastitis (high SCC) were fed with GB-9 to detect changes in milk performance, immune response, and fecal microbial community structure. The RT was used as a positive control. Results provided insight into the usage of GB-9 in reducing the usage of antibiotics in dairy farms, thereby improving food safety and public health.

## 2. Results

### 2.1. GB-9 and RT Supplementation Decreased SCC

The effects of GB-9 and RT on milk performance and composition in dairy goats were investigated. The data of milk yield, dry matter intake, milk composition were compared among the group fed with basal diet (CS), the group fed with a basal diet supplemented with 0.3% GB-9 (*w*/*w*) (BS) [12], and the group fed with basal diet supplemented with 0.3% RT (*w*/*w*) (RS) at week 0 and week 6. As shown in Figure 1, no significant differences were observed at week 0 among the three groups in milk yield, dry matter intake, and concentrations of milk protein, fat, and lactose (Figure 1A–E). Fed with GB-9 or RT did not alter milk yield, dry matter intake, and milk protein concentrations, fat, and lactose at week 6. At the beginning of the experiment (0 week), the number of SCC had a higher level in GS (*p* = 0.02) and RS (*p* = 0.01) compared with CS. After feeding with GB-9 or RT, no significant change of SCC among CS, BS, and RS was observed at week 6.

### 2.2. GB-9 and RT Altered Immune and Oxidative Stress Level in Serum

To assess the immune and oxidative stress level induced by the high level of SCC, the serum factors related to oxidative damage ((malondialdehyde (MDA), glutathione (GSH), and total antioxidant capacity (T-AOC)) were analyzed. The immune factors in serum containing immunoglobulin A (IgA), immunoglobulin G (IgG), immunoglobulin M (IgM), interleukin-2 (IL-2), interleukin-4 (IL-4), and interleukin-6 (IL-6) were also analyzed. All the selected immune and biochemical indices had no significant changes at 0 week compared with CS. The MDA concentrations in BS and RS significantly decreased compared with CS group at week 6 (Figure 2A). No significant changes in GSH and T-AOC in serum were observed after GB-9 and RT feeding (Figure 2B,C). The serum concentrations of IgA, IgM, IL-2, IL-4, and IL-6 decreased in goats treated with GB-9 or RT (Figure 2D–I).

### 2.3. GB-9 or RT Altered Fecal Microbiota Profiling

The amplicon sequencing of the 16S rRNA gene was assigned to 1937 OTUs in all fecal microbiota. The Venn diagram (Figure 3C) showed 161 unique OTUs in the BS group, 97 unique OTUs in the RS group, and 45 unique OTUs in the CS group. The three groups shared 1497 OTUs. The alpha diversity of fecal microbiota was determined, and differences were investigated by QIIME 2. The Chao 1 index in alpha diversity was significantly different between BS and CS groups, with no difference between RS and the other groups (Figure 3A,B). The PCA analysis showed a significant difference in beta diversity between BS and CS (*p* = 0.03) (Figure 3D).

The relative abundance of the top 35 genera among the CS, BS, and RS groups was used to analyze changes in bacterial profiles (Figure 4A and Appendix A). Compared with the CS group, GB-9 significantly decreased the abundance of *Bacteroides* and unidentified _*Prevotellaceae* and increased the abundance of *Akkermansia*, *Romboutsia*, *Succinivibrio*, *Butyrivibrio*, and *Phascolarctobacterium*. Compared with the CS group, the RS group had a lower abundance of unidentified *Prevotellaceae* and a higher abundance of *Succinivibrio* and *Butyrivibrio*. Further analysis using LEfSe underscored that the abundance of *Bacteroides*, *Succinivibrio*, and *Prevotellaceae* was altered significantly in GB-9 treated goats (Figure 4B).

### 2.4. Bacteroides Correlated with the Concertation of Immune Factors

Canonical correlation analysis (CCA) and redundancy analysis (RDA) proved the interaction among milk yield, milk protein, milk fat, SCC, and microbiota. Both GB-9 or RT treatment had the highest effect on the SCC, followed by MF (Appendix A). The SCC had a negative correlation with GB-9 or RT, which could be attributed to the alteration of the fecal microbiota profile.

Based on weighted gene co-expression network analysis (WGCNA), we further investigated the relationship between immune indicators and fecal microbiota. A total of 21 SCC-associated microbiota modules were identified (Figure 5A). The MEpink module with 89 interacting OTUs had significant correlation with IgA (*p* = 0.008, R = 0.48), IgG (*p* = 0.05, R = 0.37), IgM (*p* = 0.009, R = 0.48), IL-2 (*p* = 0.003, R = 0.55), IL-4 (*p* = 0.003, R = 0.54), and IL-6 (*p* = 0.02, R = 0.44). The OTUs in the MEpink module are shown in Appendix A. Network exported from the MEpink module showed that g__*Bacteroides*, o__*Bacteroidales*, s__*bacterium_P3*, and g__unidentified_*Christensenellaceae* were significantly correlated with the immunoglobulin and immune factors (Figure 5B).

## 3. Discussion

Mastitis is caused by intramammary infections and is a major problem in the dairy industry. The striking divergence in mastitis-associated intestinal microbiota is conservative among lactating mammals [2,11]. The use of probiotics to restore intestinal microbiota function is an alternative cure for mastitis [24]. In the current study, marine-derived GB-9 showed a role in mammary gland health by altering the profiling of the fecal microbiota.

*Bacillus* species have been utilized as probiotic supplements for a variety of animal species [25]. The spores of *Bacillus* species are heat-stable and can survive the low pH of the gastric barrier, so the entire dose of ingested bacteria reaches the small intestine intact [26]. Previous data in vitro showed an intrinsic character of GB-9 surviving and proliferating in culture medium with a broad range of pH [13], which helped GB-9 tolerate various pH in the rumen and gut of the ruminants. Although no direct evidence showed the colonization of GB-9 in the gut of goats, the activity of the GB-9 in the current study agreed with data in that direct-fed *Bacillus* was efficient to modify the immune system of cows preweaning calves [27,28]. The finding that no significant changes of milk yield after feeding with GB-9 was consistent with the previous data in lactating cows fed with *Bacillus pumilus* 8G-134 [28]. Although the baseline for the SCC was diverse across the treatments, the significant decrease in SCC in the BS or RS groups at 6 weeks was suggestive of a helpful role of GB-9 or RT in mammary gland health.

Antioxidant defense systems, such as MDA, GSH, and T-AOC, in metabolic tissues, can reduce the body’s oxidative stress level [29]. The minor effect of GSH and T-AOC and the lower MDA level after feeding with GB-9 or RT suggested that they have lower antioxidant potential, consistent with the lower oxidative stress level in the healthy mammary gland [30]. The immune system of mammals provides a host defense mechanism against pathogenic microbes by secreting immunoglobulins. Serum immunoglobulins play a role in resisting the invasion of bacteria and viruses, and their concentration reflects the level of the organism’s resistance to exogenous pathogens [31]. The significant decrease in the immunoglobulin level (IgA, IgG, and IgM) in the BS and RS groups might indicate a lower level of invasion of bacteria or viruses after feeding GB-9 or RT for 6 weeks.

To counteract invading pathogens, macrophages in the mammary gland initiate a defensive leucocytic response. Various defense-related components include cytokines and chemokines [e.g., IL-2, IL-4, and IL-6]. The significant increase in the IL-2 level in dairy cows suffering from mastitis [29] suggests that IL-2 is a marker of the disease. This idea is supported by the fact that IL-2 perfusion increased the SCC and IgG, IgM, and IgA concentrations to mobilize the body against the invasion of pathogens in dairy cows [32]. The IL-6 content in most cows with serious mastitis is significantly higher than that in healthy cows, suggesting its relationship with mastitis [33,34,35]. IL-6 is an early nonspecific indicator of various inflammatory states and a potential marker of subclinical inflammation [36]. In the current study, the levels of IL-2, IL-4, and IL-6 in the BS and RS groups were significantly lower than those in the CS group after 6 weeks, indicating lower immune stress in the group fed with GB-9 and RT. This fact partially agrees with the calves fed with *Bacillus subtilis natto* producing less IL-4 [27]. The results are also consistent with the fact that antibacterial mastitis therapy using ketoprofen (KET) in cows led to lower milk SCC by reducing the expression of immune factors and the inflammatory response of the mammary gland [37]. The lower level of immunoglobulins and cytokines in the group fed with GB-9 and RT indicates their role in reducing immune response, consistent with the decreased SCC. The current study results indicate that oral administration of GB-9 in lactating goats with subclinical mastitis could improve the health of the mammary gland.

The occurrence of SCC is related to the composition of the gut microbial community in sheep and cows [3,38], and thus, modifying the gut microbial community could be a new approach to reduce the risk of mastitis. In the current study, the significant change in the alpha diversity in the group fed with GB-9 compared with CS revealed the role of GB-9 in modifying the goat gut microbial community. The minimum sample of about five was accepted in the previously published data [39,40,41]. It was realized that more animals would improve the credibility of the microbiota analysis when the samples were accessible. Although the number of the animals per group was huge (the maximum was 10 in BS), our findings agreed with the previous data in lamb where GB-9 would improve the immune system by altering the profiling of the gut microbial community [12].

The role of *Phascolarctobacterium* was related to oxidative stress in sow [42]. *Bacteroides* affect the health of animals by interacting with the fecal immune system [43,44,45]. The abundance analysis showed that *Bacteroides* and *Phascolarctobacterium* were the major genera altered by GB-9 compared with RT and control. The increased abundance of *Phascolarctobacterium* in the BS group agreed with the finding in humans that prebiotic supplementation increased the abundance of this genus [46]. The lower abundance of Bacteroides and the lower immune response in the BS group were consistent with the role of Bacteroides in activating T cell-dependent immune responses [47,48]. The positive correlation between the abundance of Bacteroides and SCC and the concentrations of serum IgA, IgM, IL-2, IL-4, and IL-6 suggested that *Bacteroides* is a potential marker for diagnosis of subclinical mastitis. Although the lack of data on other factors in fecal content (for example, short-chain fatty acids), the current data suggested that the GB-9 in decreasing the SCC was associated with the alterative profile of gut microbiota.

With increasing pressure to reduce antibiotic usage, antibiotics are inhibited to treat the lactating goats without clinical signs in dairy farms to reduce antibiotic residues and antibiotic resistance. In the current study, the plant-derived RT was used as a positive control for the treatment of subclinical mastitis. The minor change in the alpha diversity in the RS group suggested that GB-9 and RT had different mechanisms in decreasing the SCC. The potential mechanism of RT decreasing SCC might be initiated from the activity compositions, including flavonoids, phenolic acids, and polysaccharides, proven to be anti-inflammatory and antioxidant agents [19,22,23]. It is realized that samples collected at serial time points will help to confirm the role of GB-9. However, with RT as a positive control, the current study suggested that administration of GB-9 had the potential to improve health in the lactating goats.

## 4. Materials and Methods

### 4.1. Experiment Design

Twenty-three subclinical mastitis dairy Saanen goats with positive CMT results (SCC > 800 × 10^4^/mL) and no clinical signs of mastitis were selected. The selected lactating goats with 110 ± 8 d in milk, milk yield of 1.2 ± 0.4 kg/d, average body weight (BW) of 52 ± 3 kg, and having lambed 3 times. All goats were from the same farm and were divided into three groups as follows: goats fed with basal diet (CS, 6 goats), goats were fed the basal diet supplemented with 0.3% GB-9 (*w*/*w*) (BS, 10 goats) [12]; and goats fed with a basal diet supplemented with 0.3% RT (*w*/*w*) (RS, 7 goats). The nutrient composition of basal diet was showen in the Appendix A. Goats were kept grouped and fed with neck clips throughout the experiment. The powder of GB-9 or RT was top-dressed onto and mixed with the TMR diet (basal diet). The goats were fed twice daily at 07:30 and 16:00 and given free access to drinking water. The experiment lasted for 7 weeks, with the first week (wk 0) for adaptation, during which the GB-9 or RT was supplemented. The nutrient compositions of the basal diet are listed in Appendix A. GB-9 (CGMCC number: 13337, accession number: CP021011) was isolated from the intestinal tract of the white-spotted bamboo shark (C. plagiosum) [49]. GB-9 powder was prepared in accordance with the reported procedure [13]. The fermentation of GB-9 was prepared following the reported procedure [12,13]. The colony-forming units in the powder were more than 2 × 10^9^/g. The RT powder was gifted from Wanshoukang Company in Jinhua, China.

### 4.2. Sample Collection and Analysis

During the treatment time, dairy goats were fed approximately 105% of their anticipated consumption. Feed and orts were weighed weekly. Dry matter intake was calculated based on the feed and orts offered and consumed. Feed samples were collected weekly and were stored at 20 °C until analysis for dry matter intake.

The data of milk yield and milk sample of each goat was collected for 3 consecutive days at 0 and 6 weeks. A milk sample was collected from individual goats following a ratio of 3:2 (morning and afternoon) [50]. The concentration of milk components (milk fat, milk protein, lactose, and SCC) was detected using CombiFoss FT + instrument (Foss Electric A/S, Hillerød, Denmark).

Blood samples were collected from all goats at 0 and 6 weeks. Blood was collected from the posterior jugular vein of each goat from 9:00 a.m. to 10:00 a.m. The serum was separated by centrifugation. The serum levels of MDA, GSH, T-AOC, IgA, IgG, IgM, IL-2, IL-4, and IL-6 were measured using specific commercial kits (Nanjing Jiancheng Biotech, Nanjing, Jiangsu, China) and semiautomatic equipment (BioPlus 2000, Kolding, Denmark) [24,29].

Feces were collected at the end of the experiment (6 weeks). The homogenized sample was snap-frozen in liquid nitrogen and stored at −80 °C for subsequent DNA analysis.

### 4.3. 16S rRNA Gene Sequencing

The samples’ total genome DNA was extracted using a commercial kit (Tiangen Biotech, Beijing, China). DNA concentration and purity were monitored on 1% agarose gels. The V3–V4 regions of the bacterial 16S rRNA genes were amplified, sequenced, and paired-end sequenced (2 × 300 bp) on the Illumina MiSeq platform following the standard protocols (Novogene Technology Co. Ltd., Tianjing, China).

### 4.4. Sequencing Data Analysis

Raw reads of different samples were demultiplexed and quality-filtered following previous methods [38]. Bioinformatics was conducted using the QIIME 2. Shannon and Chao1 indices were used to estimate bacterial richness and community diversity [51]. PCA was applied to visualize the dissimilarity of microbial communities among different age groups. LEfSe analysis was performed online (http://huttenhower.sph.harvard.edu/galaxy/root?tool_id=testtoolshed.g2.bx.psu.edu%2Frepos%2Fgeorge-weingart%2Flefse%-2FLEfSe_for%2F1.0, accessed on 18 July 2021). CCA and RDA were performed at the genus level by using the vegan package in R. Correlation heatmaps were generated using the R program pheatmap package. WGCNA analysis was performed using the WGCNA package in R, and the network was drawn by Cytoscape (Version 3.7.1).

### 4.5. Statistical Analysis

The data of milk performance and composition were analyzed using the one-way ANOVA with the SPSS software (SPSS v.19, SPSS Inc., Chicago, IL, USA). The statistical analyses of serum factors and microbiol diversity were performed by one-way ANOVA. Data are presented as means plus SEM. *p* < 0.05 was considered statistically significant. Correlation networks were generated using Spearman’s rank correlation coefficients and visualized using the Cytoscape. The significant correlation between bacterial genus and the immune globulins and cytokines was considered when |R| > 0.2 and *p* < 0.05).

## 5. Conclusions

With the increasing pressure on antibiotics usage, probiotics support is an ideal strategy for controlling mastitis in dairy farms. Feeding GB-9 and RT significantly reduced the SCC in dairy goats with subclinical mastitis. The lower concentration of immune and biochemical factors in serum after feeding with GB-9 indicated that this marine-derived probiotic could be used to improve the health of the mammary gland. The alteration of immune and biochemical factors in the group fed with Gb-9 was associated with gut microbiota profiling changes. *Bacteroides* and *Phascolarctobacterium* were the major genera response to GB-9 feeding and were positively correlated with the concentrations of IgA, IgM, IL-2, IL-4, and IL-6. Future studies are needed to identify the mechanism of GB-9 in altering fecal microbiota composition and the immune system. However, future work on the colonization and establishment of GB-9 in the gut will help understand the crosstalk between the gut microbiota (e.g., *Bacteroides*) and immune system. Collectively, the current data revealed that GB-9 could be a helpful probiotic to control subclinical mastitis.

## Figures and Tables

**Figure 1 marinedrugs-19-00404-f001:**
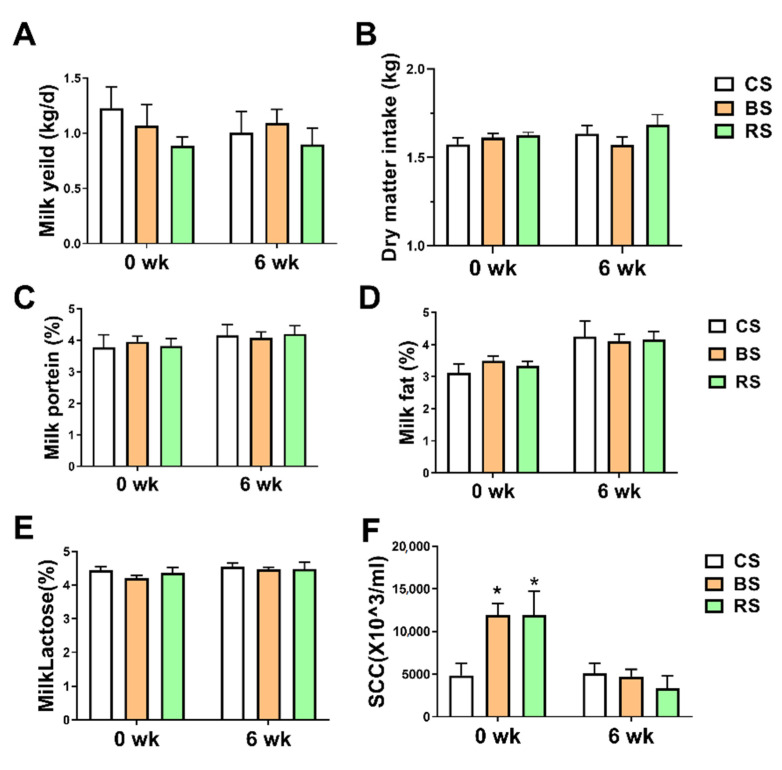
The effect of the administration of Bacillus amyloliquefaciens-9 on milk performance and somatic cell count (SCC) in three groups: fed with basal diet (CS) basal diet with 0.3% (*w*/*w*) Bacillus amyloliquefaciens-9 (GB-9) (BS), and basal diet with 0.3% (*w*/*w*) Radix Tetrastigmae powder (RS) at week 0 and week 6. (**A**) Milk yield. (**B**) Dry matter intake. (**C**) The concentrations of milk protein. (**D**) The concentrations of milk fat. (**E**) The concentrations of milk lactose. (**F**) The value of SCC. Statistical analyses were performed using the ANOVA with the SPSS software (SPSS v.19, SPSS Inc., Chicago, IL, USA). Data were presented as mean ± SEM. * *p* < 0.05 was considered statistically significant compared with CS group.

**Figure 2 marinedrugs-19-00404-f002:**
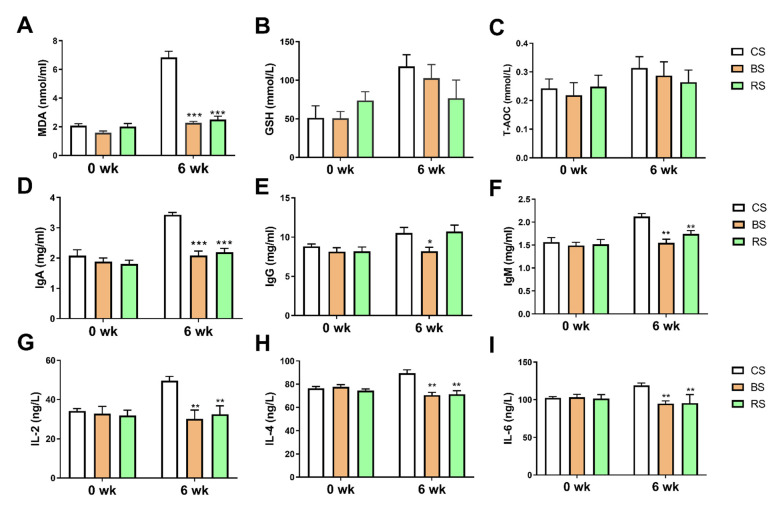
The effect of the administration of *Bacillus amyloliquefaciens-9* on serum immune and biochemical indices of three groups: fed with basal diet (CS) basal diet with 0.3% (*w*/*w*) *Bacillus amyloliquefaciens*-9 (GB-9) (BS), and basal diet with 0.3% (*w*/*w*) *Radix Tetrastigmae* powder (RS) at week 0 and week 6. (**A**) MDA concentrations in serum. (**B**) GSH concentrations in serum. (**C**) Total antioxidant capacity of serum. (**D**) Serum IgA concentrations. (**E**) Serum IgG concentrations. (**F**) Serum IgM concentrations. (**G**) Serum IL-2 concentrations. (**H**) Serum IL-4 concentrations. (**I**) Serum IL-6 concentrations. Data are mean ± SEM. Statistical analyses were performed using the ANOVA with the SPSS software (SPSS v.19, SPSS Inc., Chicago, IL, USA). * *p* < 0.05, ** *p* < 0.01 and *** *p* < 0.005 compared with the CS group.

**Figure 3 marinedrugs-19-00404-f003:**
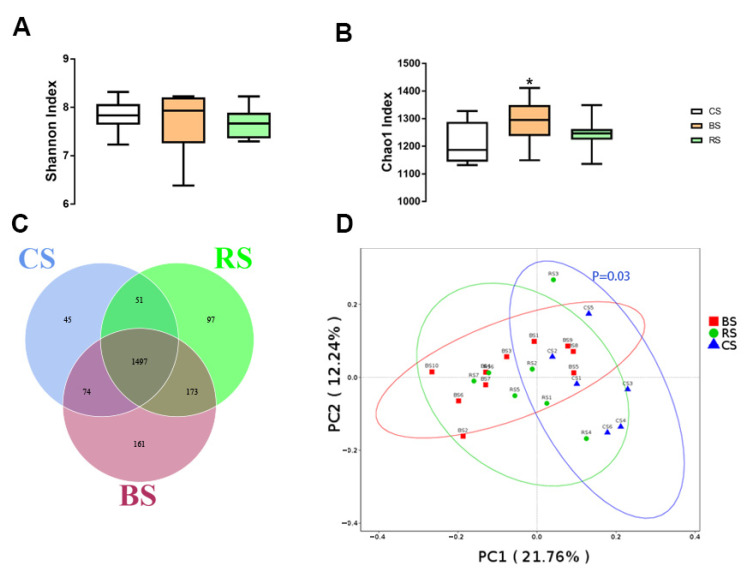
Fecal microbiota characteristics in three groups: fed with basal diet (CS) basal diet with 0.3% (*w*/*w*) *Bacillus amyloliquefaciens*-9 (GB-9) (BS), and basal diet with 0.3% (*w*/*w*) *Radix Tetrastigmae* powder (RS) at week 6. Fecal samples were collected at week 6. Bacterial 16S rRNA genes were amplified and sequenced. (**A**) Alpha diversity (Shannon index). (**B**) Alpha diversity (Chao 1 index). (**C**) OTU Venn diagram. (**D**) PCA analysis. Data are mean ± SEM. * *p* < 0.05. Statistical analyses were performed using the ANOVA with the SPSS software (SPSS v.19, SPSS Inc., Chicago, IL, USA). * *p* < 0.05 was considered statistically significant compared with the CS group.

**Figure 4 marinedrugs-19-00404-f004:**
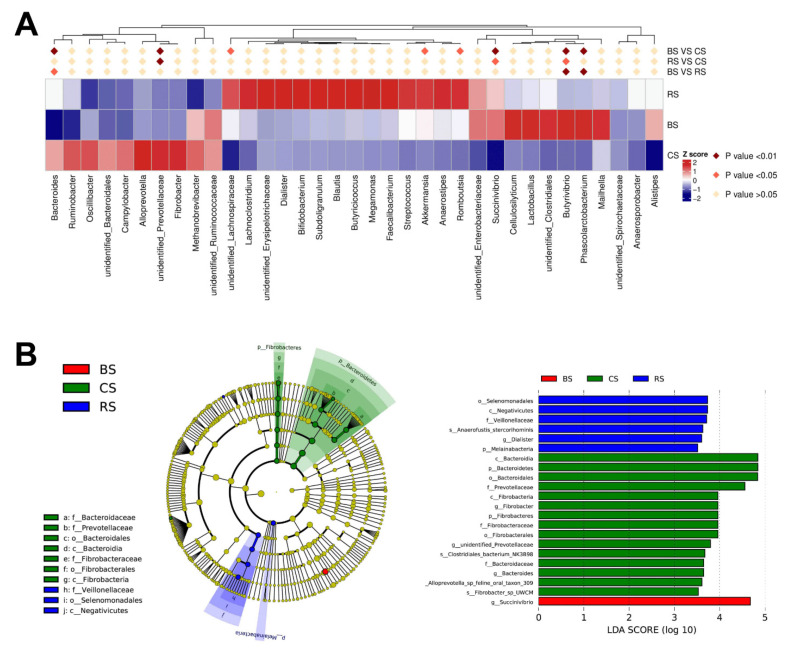
Differences in microbiota abundance in the three groups: fed with basal diet (CS) basal diet with 0.3% (*w*/*w*) Bacillus amyloliquefaciens-9 (GB-9) (BS), and basal diet with 0.3% (*w*/*w*) *Radix Tetrastigmae* powder (RS) at week 6. (**A**) The heatmap of the top 35 genera. (**B**) The result of LEfSe. Bar plots show linear discriminant analysis (LDA) scores of each OTU. The LDA score indicates the effect size of each OTU, and OTUs with an LDA score > 3.0 are shown. Statistical analyses of the relative microbial abundance in genus level were performed using the ANOVA with the SPSS software (SPSS v.19, SPSS Inc., Chicago, IL, USA).

**Figure 5 marinedrugs-19-00404-f005:**
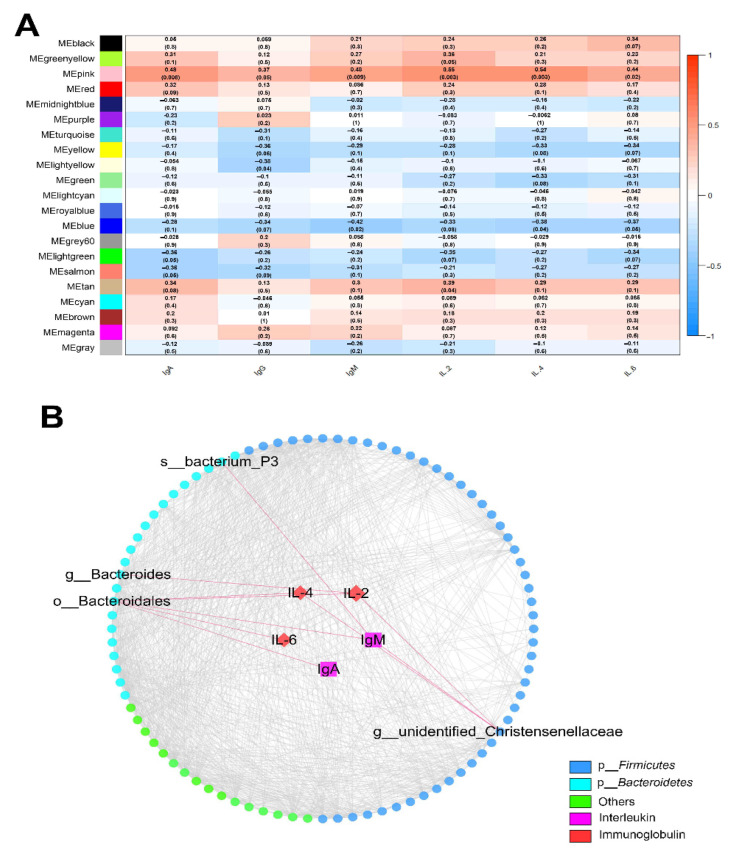
Correlation between the concentrations of immune indicators and abundance of fecal microbiota. The correlated module was analyzed using a weighted gene co-expression network analysis (WGCNA). Correlation networks were generated using Spearman’s rank correlation coefficients and visualized using the Cytoscape. (**A**) The heatmap of the WGCNA module. (**B**) Correlation network among immunoglobulins, cytokines, and 89 interacting OTUs in MEpink module. Strong correlations between OTUs and immune index (|R| ≥ 0.5 and *p* < 0.05) are displayed in the network in the red edge.

## Data Availability

The identified sequences were deposited in the NCBI, and the accession number in NCBI BioProject was PRJNA664921.

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
