# Peer review of "Bacillus amyloliquefaciens*-*9* Reduces Somatic Cell Count and Modifies Fecal Microbiota in Lactating Goats"

_marinedrugs, 2021, doi:10.3390/md19080404_

Round 1
Reviewer 1 Report
The manuscript describes the association between the administration of Bacillus amyloliquefaciens-9/Radix Tetrastigmae powder and selected immunological/ production parameters in dairy goats.
In the introduction section, authors highlighted the importance of subclinical mastitis and presented basic data regarding the origin of Bacillus amyloliquefaciens and Radix Tetrastigmae as well as their properties observed in similar researches. The aim of the research is clearly stated.
The material and method section requires more detailed explanation.
Authors are asked to explain whether animals enrolled in the research were chosen from the same farm or from different farms.
Authors stated (line 263) “Goats were kept were separately throughout the experiment.“ It is not clear whether animals were kept individually or grouped.
Furthermore, authors stated (line 264) „The powder of GB-9 or RT were top-dressed onto TMR diet (basal diet).“ If animals were grouped, are authors confident that each animal in a group consumed sufficient quantity of the powder?
It is not clear why authors analysed pooled milk samples rather than morning and afternoon samples separately since analysis is quite cheap and fast.
Finally, authors stated (line 260/263) „All goats were randomly divided into three groups…“ However it seems that control group differs from experimental groups significantly since the somatic cell count is approximately doubled in experimental groups compared with the control and groups are unequal by size. Authors are asked to explain.
The paired t-test should be used to compare longitudinal data rather than t-test for independent samples.
Authors applied different approach in the statistical analysis between production and immunological parameters. Production parameters were compared within a same group (start vs. termination of the research) while immunological parameters were compared between groups at the start and termination.
In the discussion section authors compared their own results with the results of similar studies. However some explanations of the differences between control and experimental groups are questionable. For example authors stated (line 204/205) “The finding is consistent with fact that the mammary gland with subclinical mastitis has high levels of serum immunoglobulins“. This statement would be convincing if authors provided referent values for parameters or by inclusion of healthy control group. Without referent values or healthy control group, an alternative explanation for lower values of immunological parameters in treated animals could be the immunosuppression. Even if the above authors’ statement is correct ie if used supplements modulate immunological response, authors should exclude concurrent infections at other body sites that could have the influence on the immunological parameters. The cited sentence relies on the immunoglobulins level in mammary gland while authors determined this parameter in blood sample.
Overall the manuscript presents very interesting data that could improve food safety and public health.
Supplementary files cannot be seen since the error message appears after clicking the link.
Author Response
Dear editor and reviewers,
We really appreciate all your comments and suggestions! Please find my itemized responses in below and my corrections in the re-submitted files. We also corrected the English writing throughout the text and marked in yellow.
Thanks again!
Reviewer #1
Comments and Suggestions for Authors
- The manuscript describes the association between the administration of Bacillus amyloliquefaciens-9/Radix Tetrastigmae powder and selected immunological/ production parameters in dairy goats.
Respond: Thanks for the positive comment.
- In the introduction section, authors highlighted the importance of subclinical mastitis and presented basic data regarding the origin of Bacillus amyloliquefaciens and Radix Tetrastigmae as well as their properties observed in similar researches. The aim of the research is clearly stated.
Respond: We appreciate the positive comment.
- The material and method section requires more detailed explanation.
Respond: Thanks for the comment. We added the detail of the experiment design in the material and method section. The sentences were marked in yellow. (Line 258, 261-263, 279 and 306-307)
- Authors are asked to explain whether animals enrolled in the research were chosen from the same farm or from different farms.
Respond: Thanks for the comment. They were chosen from the same farm. We added the statement in the section. (Line 258-259)
- Authors stated (line 263) “Goats were kept were separately throughout the experiment.“ It is not clear whether animals were kept individually or grouped.
Respond: Thanks for the comment. They were kept grouped. The sentence was rewritten. (Line 261-263)
- Furthermore, authors stated (line 264), The powder of GB-9 or RT were top-dressed onto TMR diet (basal diet).“ If animals were grouped, are authors confident that each animal in a group consumed sufficient quantity of the powder?
Respond: We apologize for the unclear statement. Firstly, the powder were mixed with basal diet at w/w. Secondly, the each animal was fed with neck clip and it helped the goat consumed anticipated consumption. The statement was inserted in Line 261-263.
- It is not clear why authors analysed pooled milk samples rather than morning and afternoon samples separately since analysis is quite cheap and fast.
Respond: The concentration of milk components is commonly analyzed through the pooled milk due to the inhomogeneity of milk at each milk. The reference was inserted in Line279.
- Finally, authors stated (line 260/263), All goats were randomly divided into three groups…“ However it seems that control group differs from experimental groups significantly since the somatic cell count is approximately doubled in experimental groups compared with the control and groups are unequal by size. Authors are asked to explain.
Respond: Sorry for the unclear statement. The sentence was rewritten according to the comment. (Line 258)
- The paired t-test should be used to compare longitudinal data rather than t-test for independent samples.
Respond: Thanks for the comment. The approach in the statistical analysis of milk components was changed to comparison between groups at the start and termination according to the comment. The sentences in results and statistical analysis section were rewritten. (Line 75-84, 91-92 and 306-307)
- Authors applied different approach in the statistical analysis between production and immunological parameters. Production parameters were compared within a same group (start vs. termination of the research) while immunological parameters were compared between groups at the start and termination.
Respond: Thanks for the comment. The approach in the statistical analysis of milk components was changed to comparison between groups at the start and termination according to the comment. The sentences in results and statistical analysis section were rewritten. (Line 75-84, 91-92 and 306-307)
- In the discussion section authors compared their own results with the results of similar studies. However some explanations of the differences between control and experimental groups are questionable. For example authors stated (line 204/205) “The finding is consistent with fact that the mammary gland with subclinical mastitis has high levels of serum immunoglobulins“. This statement would be convincing if authors provided referent values for parameters or by inclusion of healthy control group. Without referent values or healthy control group, an alternative explanation for lower values of immunological parameters in treated animals could be the immunosuppression. Even if the above authors’ statement is correct ie if used supplements modulate immunological response, authors should exclude concurrent infections at other body sites that could have the influence on the immunological parameters. The cited sentence relies on the immunoglobulins level in mammary gland while authors determined this parameter in blood sample.
Respond: Thanks for the comment. This sentence was removed. We also checked and corrected the sentences throughout the discussion section.
- Overall the manuscript presents very interesting data that could improve food safety and public health.
Respond: We appreciate the positive comment.
- Supplementary files cannot be seen since the error message appears after clicking the link.
Respond: Sorry for the error. We will double check when the revised manuscript is uploaded.

Reviewer 2 Report
The manuscript by Li et al. demonstrates the ability of the probiotic Bacillus amyloliquefaciens-9 to control subclinical mastitis in dairy goats.
Changes in milk performance, immune response and fecal microbial composition have been shown.
The impact is in the potential of probiotic-based strategies to reduce antibiotic usage in dairy farms, exploiting the gut-brain axis.
Some points need to be addressed:
- The number of goats per group should be increased: 10 animals per group are required for microbiota analyses.
- The authors used Radix Tetrastigmae as anti-inflammatory positive control. In the reviewer opinion the authors should also compare the effect of Bacillus amyloliquefaciens-9 with the classical antibiotic therapy. By adding this not-negligible control many interesting data could be highlighted.
- Short chain fatty acids fecal content should be measured to support the hypothesis that anti-inflammatory effect is exerted by modifying gut microbiota composition.
- Paragraph 2.2 contains several grammar and misspelling errors.
Author Response
Dear editor and reviewer,
We really appreciate all your comments and suggestions! Please find my itemized responses in below and my corrections in the re-submitted files. We also corrected the English writing throughout the text and marked in yellow.
Thanks again!
Reviewer #2
Comments and Suggestions for Authors
- The manuscript by Li et al. demonstrates the ability of the probiotic Bacillus amyloliquefaciens-9 to control subclinical mastitis in dairy goats. Changes in milk performance, immune response and fecal microbial composition have been shown. The impact is in the potential of probiotic-based strategies to reduce antibiotic usage in dairy farms, exploiting the gut-brain axis.
Respond: We appreciate the positive comment.
- The number of goats per group should be increased: 10 animals per group are required for microbiota analyses.
Respond: Thanks for the comment. It is realized that the problem might influence resolution of the microbiota community. We stated the limitation in the conclusion section. (Line 323-324)
- The authors used Radix Tetrastigmae as anti-inflammatory positive control. In the reviewer opinion the authors should also compare the effect of Bacillus amyloliquefaciens-9 with the classical antibiotic therapy. By adding this not-negligible control many interesting data could be highlighted.
Respond: Thanks for the comment. We agree with the comment that classical antibiotic therapy will help highlight the role of GB-9 in anti-inflammatory. However, the antibiotic is too strong to alter the microbiota community of animals. To addition, antibiotic therapy significantly reduces the risk of having mastitis but may lead to the decline of dairy cow quality and formation of antibiotic residues and antibiotic resistance.
- Short chain fatty acids fecal content should be measured to support the hypothesis that anti-inflammatory effect is exerted by modifying gut microbiota composition.
Respond: Thanks for the comment. We inserted a discussion and stated the limitation of the current study in the discussion section. Although lack of the data of short chain fatty acids in fecal content in the present study, our findings still suggested that the mechanism of GB-9 in decreasing the SCC in-volves altering the profile of gut microbiota, (Line 241-242)
- Paragraph 2.2 contains several grammar and misspelling errors.
Respond: we appreciate this comment. The errors were double checked and corrected throughout the text.

Round 2
Reviewer 1 Report
Authors accepted all reviewer's suggestions hence no further requests. Please check line 91: ANOVN or ANOVA?
Author Response
1.Please check line 91: ANOVN or ANOVA?
Respond: It was corrected to ANOVA. The erorr was corrected throughout the text.
Reviewer 2 Report
The authors didn’t consider at all the reviewer comments.
In a peer-reviewed paper an adequate sample size and a good experimental design including all the needed controls are imperative. Considering these unresolved issues, the paper is not suitable for publication.
